# Incremental Ordering for Scheduling Problems

**Primary Keywords:** *(4) Theory;*

## Abstract

Given an instance of a scheduling problem where we want to start executing jobs as soon as possible, it is advantageous if a scheduling algorithm emits the first parts of its solution early, in particular before the algorithm completes its work. Therefore, in this position paper, we analyze core scheduling problems in regards to their *enumeration complexity*, i. e. the computation time to the first emitted schedule entry (*preprocessing time*) and the worst case time between two consecutive parts of the solution (*delay*).

Specifically, we look at scheduling instances that reduce to ordering problems. We apply a known incremental sorting algorithm for scheduling strategies that are at their core comparison-based sorting algorithms and translate corresponding upper and lower complexity bounds to the scheduling setting. For instances with $n$ jobs and a precedence DAG with maximum degree $\Delta$, we incrementally build a topological ordering with $O(n)$ preprocessing and $O(\Delta)$ delay. We prove a matching lower bound and show with an adversary argument that the delay lower bound holds even in case the DAG has constant average degree and the ordering is emitted out-of-order in the form of insert operations.

We complement our theoretical results with experiments that highlight the improved time-to-first-output and discuss research opportunities for similar incremental approaches for other scheduling problems.

## 1   Introduction

Assigning resources to work packages ("jobs") in a way that optimizes an objective is a ubiquitous task in most industrial settings. Naturally, it has been a scientific endeavor for many years to develop efficient algorithms and to understand the limits of possible algorithmic solutions by proving lower bounds on the time complexity of such scheduling problems (Pinedo 2022; Brucker et al. 2023; Brucker 2007).

Most research has focused on a *total-time* complexity perspective: An algorithm is presented with an input instance to a scheduling problem, and the time complexity is analyzed (or measured in case of experimental results) from the first computation steps until the algorithm outputs the complete solution. This approach makes it possible to employ complex algorithmic strategies, but it also means that we cannot start to execute the schedule until the scheduling algorithm terminates.

In contrast to this, we propose to analyze scheduling problems with an *enumeration* perspective: After some initial *preprocessing time*, an enumeration algorithm has to emit parts of the final solution with little *delay* in between consecutive solutions. In the case of scheduling algorithms, these solution parts are schedule entries that specify when and on which resource to execute a job. An algorithm that enumerates the first entries of a schedule early on enables us to start executing the schedule before it is finalized.

Note that this is different from the well-known *online* setting: There, an algorithm does not see the whole input instance but is presented with the jobs to be scheduled one at a time and has to make an immediate scheduling decision (Albers 2009). This limitation is also present in both the *semi-online* setting and in its generalization, *scheduling with advice*, where the algorithm has access to additional information on the input sequence or output properties (Dwibedy and Mohanty 2022; Boyar et al. 2016). In contrast, our enumeration algorithms do have access to the complete input and are free to decide which jobs to schedule next.

Finally, we want to highlight that enumerating solution parts gives a head start to subsequent steps in a processing pipeline. Therefore, the approach can be advantageous even when the resulting total time of the enumeration algorithm is worse than a classical total-time algorithm, as Lindner et al. (2017) showed for a DNA sequencing application.

### 1.1   Our Contribution

Many basic scheduling variants can be solved by ordering jobs according to their precedence graph and/or sorting available jobs according to a simple scheduling rule.

After formally introducing the concept of an enumeration algorithm and basic notation in section 2 we tackle topological orderings of precedence graphs in section 3. We discuss why enumerating the solution in ascending order requires known in-degrees and preprocessing time linear in the number of vertices. Regarding delay, we prove that enumerating a topological ordering of a graph with maximum out-degree $\Delta^+$ both requires and is possible with $\Theta(\Delta^+)$ delay.

In section 4 we apply both our algorithm for ascending topological ordering enumeration as well as INCREMEN­TALQUICKSELECT by Paredes and Navarro (2006) for incremental sorting (see related work) to scheduling problems. With these two algorithms we compute schedules that opti-

mize the maximum completion time ("makespan") in three different settings (single machine with either release times or a precedence graph, flow shop with two machines) and discuss how the results transfer to other scheduling variants. We demonstrate the improved time-to-first-output with experiments on random instances of the three main settings.

As this position paper intends to spark further work on schedule enumeration, we conclude in section 5 with a number of potential research directions.

## 1.2 Related Work

The proposed enumeration perspective has already been studied for core algorithmic problems on graphs and (un)ordered sets:

When computing *Single Source / All Pairs Shortest Distances* in graphs, the order of enumerated solution parts plays a crucial role in the enumeration complexity (Casel et al. 2024). Our lower bound on the delay of topological ordering follows a similar approach as in this work. We will, however, show that the order of enumerated parts does not influence the complexity of topological ordering enumeration.

*Incremental Sorting* is concerned with emitting the elements of an input set in ascending order. Paredes and Navarro (2006; 2010) developed the INCREMENTALQUICKSELECT algorithm to solve this problem with optimal preprocessing and delay and showed its practical use in an experimental application to computing a minimum spanning tree. We apply this algorithm to enumerating schedules to problems where sorting according to a simple rule yields an optimal solution.

Carmeli et al. (2022, Proposition 3.9) devised an enumeration variant of the *Fisher-Yates Shuffle* (Durstenfeld 1964) to derive random-permutation enumeration algorithms from random-access enumeration algorithms in the field of database queries.

## 2 Enumeration Model and Notation

In the standard *total-time* model, an algorithm maps an *input* to a *solution* that has to fulfill problem-specific correctness and optimality criteria. Time complexity is analyzed/measured as the *total time* from the start (input is provided) to the termination (algorithm produces output).

**Enumeration Problems** We study algorithms that *enumerate* parts to a solution: Enumeration algorithms are provided with an *input*, perform initial computations and prepare data structures in a *preprocessing phase*, and then, in the *enumeration phase*, emit a number of *solution parts* without repetition.

The specific type and semantics of a solution part are specified in a problem description, along with an optional requirement on the order in which the parts have to be produced. It is, however, a general requirement that the emitted parts can be efficiently assembled to a complete solution that has to fulfill the same criteria as in the standard model.

Time complexity is analyzed/measured in terms of the *preprocessing time* the algorithm spends in the preprocessing phase and the worst-case *delay* the algorithm spends in

the enumeration phase before emitting the first / next solution part.

In this work we consider three types of enumeration problems:

**Topological Ordering Enumeration** The input is a directed acyclic graph (DAG) $G = (V, E)$ with $n$ vertices $V = [n] = \{1, 2, \ldots, n\}$. For a vertex $v$ we write $\delta^+(v)$ for the out-degree of $v$. We denote by $\Delta^+ = \max_{v \in V} \delta^+(v)$ the maximum out-degree of the graph. A topological ordering for $G$ is a permutation $\pi$ of its vertices, formalized as tuple $(\pi(1), \pi(2), \ldots, \pi(n))$, such that for all edges $(u, v) \in E$ it holds that $\pi^{-1}(u) < \pi^{-1}(v)$.

An algorithm that solves the *ascending topological ordering enumeration* problem has to produce as solution parts the individual entries $\pi(1), \pi(2), \ldots, \pi(n)$ in that order.

**Incremental Sorting** The input is a set $X$ of elements with a total order $\preceq$, the solution is a permutation $\pi$ that is sorted in ascending order according to $\preceq$.

An enumeration algorithm for this problem has to produce the individual permutation entries $\pi(1), \pi(2), \ldots, \pi(n)$ in the sorted order.

**Schedule Enumeration** Because scheduling of jobs happens in many different environments there are numerous problem variants studied in the literature. Many of them are commonly named according to a three field classification scheme $\alpha|\beta|\gamma$ for the machine environment, problem characterics and objective function (Graham et al. 1979). We summarize here the notions required for this paper and refer for a more in-depth introduction to standard literature on the topic (Brucker 2007; Pinedo 2022). In section 4, when we apply our techniques, we will introduce the specific problem definitions.

A scheduling input instance consists of $n$ jobs identified by $j \in [n]$ to be scheduled for processing in a machine environment $\alpha$, e. g. on a single machine ($\alpha = 1$) or in a flow shop on machine 1 first and then on machine 2 ($\alpha = F2$).

Each job $j$ consists of one or more operations $o$, each with a processing time $p_{j,o}$. A job can be processed on only one machine at a time, no machine can process multiple jobs simultaneously and we only consider the case without preemption (processing an operation cannot be interrupted).

Additional information and/or requirements $\beta$ might be provided. Examples include release times $r_j$ and deadlines $d_j$ per job, or a precedence relation given as DAG where the edge $(j, k)$ specifies that job $j$ has to be completed before work on job $k$ can start.

A scheduling algorithm has to produce a schedule, i. e. an assignment of jobs to processing intervals on machines. The schedule has to optimize some objective function $\gamma$, e. g. minimize $C_{\max}$, the completion time of the last job.

Enumerating such a schedule means emitting entries of the form $(j, i, s)$ that specify that job $j$ is to be executed on machine $i$ in the time interval starting at time $s$. (We omit $i$ in the single machine setting.) In this work we will consider the most natural version in which the entries have to be enumerated in the order of increasing start time.

# 3 Topological Ordering Enumeration

Recall that for *ascending topological ordering enumeration* an algorithm has to enumerate a topological ordering from $\pi(1)$ to $\pi(n)$. Especially regarding the applications to scheduling that we have in mind, enumerating a topological ordering in that way appears to be the most natural.

However, this restriction on the output order has implications on the complexity of the enumeration task at hand. If the input is a graph in the form of out-adjacency lists without information on the in-degrees of vertices, one clearly needs $\Omega(n + m)$ time in the worst case to identify a vertex without in-coming arcs (*source vertex*) to be enumerated as $\pi(1)$. Given that computing a complete topological order via finish times of a depth first search is in $O(n + m)$, this rules out enumeration as a suitable tool for this setting. We will, therefore, assume that each vertex in our input data knows its in-degree, for example by having both in- and out-adjacency lists with a size attribute per list. Without further information, we certainly still need $\Omega(n)$ preprocessing time to identify source vertices.

**Corollary 1.** *Ascending topological ordering enumeration without in-degree information needs preprocessing time in $\Omega(n + m)$. With known in-degrees but unknown source vertices preprocessing time is in $\Omega(n)$.*

Once all source vertices are known (either by $O(n)$ preprocessing or by assuming that they are given additionally as input), the classical iterative source removal algorithm can enumerate a topological ordering with delay in $O(\Delta^+)$: The algorithm stores all current source vertices in a queue and keeps track of the in-degree of all visited vertices in an array $D$. In the enumeration phase, the algorithm repeats the following until the queue is empty: It removes the first vertex $u$ from the queue. For each of $u$'s outgoing edges $(u, v)$ it decrements the in-degree of vertex $v$ and appends $v$ to the queue of source vertices should $v$'s degree reach 0. Then, the algorithm emits $u$ as solution part after $O(\delta^+) \subseteq O(\Delta^+)$ steps.

**Corollary 2.** *Ascending topological ordering enumeration with known in-degrees can be solved with preprocessing time in $O(n)$, delay in $O(\Delta^+)$ and space-complexity in $\Theta(n)$. If the set of source-vertices is given as input, the preprocessing time can be reduced to $O(1)$ using lazy-initialized memory.*[1]

The preprocessing for this enumeration is certainly optimal, but it is not apparent why the delay needs to be $O(\Delta^+)$. Considering that $O(n + m)$ suffices to compute a whole topological ordering and $n$ solution parts are produced, one could hope to improve the delay to the average degree of $G$. To properly study the possibility of such improvement, we consider a variation of the enumeration problem that does not require preprocessing. This allows to highlight the worst-case delay. For this version, we will show that delay in $O(\Delta^+)$ is optimal, and then discuss resulting implications for ascending enumeration.

---

[1]For a detailed discussion of memory considerations in enumeration see (Casel et al. 2024).

For *topological ordering enumeration* (without the addition of *ascending*), we consider solution parts to be insertion operations $y_1, y_2, \ldots, y_n$ where each $y_i$ has the form $(v, p) \in V \times (V \cup \{\varepsilon\})$: "In step $i$, insert vertex $v$ (a) if $p = \varepsilon$ at the beginning of the current partial ordering or (b) else after vertex $p$.". We assume that the ordering is to be written from left to right, thus *after* means *right of*. Insertion operations have to be feasible, meaning that every vertex is inserted exactly once, and that instructions to insert after some vertex $p$ are preceded by instructions to insert vertex $p$. Formally, any two instructions $y_i = (p, \cdot), y_j = (\cdot, p)$ have to fulfill $i < j$.

Note that this is another way of enumerating $n$ parts that can be used to construct a topological ordering and that this way is a generalization of the *ascending* enumeration introduced before. The ascending restriction only adds that each insertion happens at the end of the current partial order; thus for all $2 \leq i \leq n$ with $y_i = (\cdot, p)$ the previous operation must be $y_{i-1}(p, \cdot)$.

For the generalized notion of topological ordering enumeration, a repeated depth-first search can be used to enumerate with delay in $O(\Delta^+)$ without preprocessing or additional knowledge about the input.

**Theorem 3.** *Topological ordering enumeration can be solved with delay in $O(\Delta^+)$ with $\Theta(n)$ lazy-initialized memory and space-complexity in $\Theta(n)$.*

*Proof.* The algorithm tracks in a lazy-initialized boolean array $A$ of length $n$ in field $A[u]$ whether vertex $u$ was visited by one of the searches. Uninitialized entries are read as FALSE. It further maintains a queue $Q$ to collect solution parts for later output. This queue is filled as shown in Algorithm 1.

---

**Algorithm 1:** Topological ordering enumeration

---

1 **foreach** $v = 1$ **to** $n$ **do**
2     **if** $A[v] \neq$ TRUE **then**
3        visit$(v, \varepsilon)$;

4 **Function** visit$(v, parent)$:
5     enqueue$(v, parent)$;
6     $A[v] =$ TRUE;
7     **forall** $(v, w) \in E$ **do**
8        **if** $A[w] \neq$ TRUE **then**
9           visit$(w, v)$;

---

Note that Algorithm 1 is essentially a standard depth first search augmented by line 5 that produces the solution parts and fills $Q$. This queue will now be used by our enumeration algorithm to emit solution parts while minimizing the worst case delay.

Let $c$ be some implementation specific constant. After emitting a solution part $(v, parent)$, the algorithm delays the output of the next solution part from $Q$ by $c \cdot \delta^+(v)$ steps. Whenever the algorithm is required to emit an output, $Q$ must not be empty. We apply the accounting method to prove this. For each credit unit, the algorithm can perform

a constant number of steps. As long as the credit stays positive, $Q$ is not empty.

Initially, the credit balance is $\Delta^+$, as the delay we want to prove is in $O(\Delta^+)$. This pays for all computation steps in the first iteration of the loop in line 1 until the first solution part is produced. Each time a solution part $(v, parent)$ is enqueued in line 9, this part is charged $\delta^+(v) \in O(\Delta^+)$ credit. This credit pays for all computation in the current recursive invocation of `visit` and, (a) in case $parent \neq \varepsilon$ for skipping $v$ in a later iteration of the loop in line 1, (b) in case $parent = \varepsilon$ for the next iteration of the same loop that discovers an unvisited vertex. This implies that solution part pays for all computation associated to the visited vertex, including checking its immediate descendants and backtracking the DFS. As each invocation of `visit` immediately enqueues a new solution part, the credit therefore stays positive.

Hence, the algorithm enumerates solution parts with delay in $O(\Delta^+)$.

Whilst computing a topological ordering by running DFS on all unvisited vertices is a standard algorithm, usually each vertex is inserted at the head of the ordering as soon as the search fully processed all its descendants and backtracks from them. Our Algorithm 1 deviates from that by inserting vertices immediately when they are visited. However, this does not change the main property of the DFS-based topological ordering: All descendants of a vertex are placed to the right of it. We prove correctness of Algorithm 1 by showing that all edges are forward edges in the joined ordering.

First note that root nodes of depth-first searches are always inserted at the head of the ordering (cp. line 3). Secondly observe that, except for these root nodes, all other vertices are inserted directly after their parent in the DFS tree (cp. line 9). Therefore, whenever a call to `visit` on a vertex $v$ ends and the search backtracks, no further insertions will happen after $v$'s position in the current partial ordering.

For each edge $(u, v) \in E$ there are two possible cases:

1. Vertex $u$ is inserted into the ordering first. This implies that $v$ is visited as a descendant of $u$ by the depth-first search and therefore inserted directly after $u$ or after one of $u$'s other descendants. Therefore, $(u, v)$ is a forward edge.
2. Vertex $v$ is inserted into the ordering first. As $(u, v) \in E$ implies that $u$ is not a descendant of $v$, the search completely backtracks to a parent of $v$ or even to the root loop in line 1 and thus, according to the earlier observation, $u$ is later inserted left of $v$ and $(u, v)$ is a forward edge.

Thus, the algorithm produces a correct topological ordering of the input DAG. □

This positive result can be matched with a corresponding lower bound by creating an adversarial input. For this, consider the graph structure in Figure 1: For given $k$, the graph consists of $k$ vertices that form a fully connected DAG $C$, a set $B$ of $k - 1$ vertices that each add a bridge between a pair of two consecutive vertices from $C$, and a path $P$ of $k^2$ vertices that extends one of those bridges. The idea of the

lower bound proof is now to force any solution algorithm to essentially fully process the fully connected DAG before it can figure out, where the path of $k^2$ vertices appears in the ordering.

**Theorem 4.** *Topological ordering enumeration cannot be solved with delay in $o(\Delta^+)$, even if the graph has constant average out-degree.*

*Proof.* Assume some algorithm $A$ was able to enumerate the solution parts with delay in $o(\Delta^+)$ and consider the following adversarial setup that is equivalent to receiving the input graph as adjacency lists with attached in-degree information: $A$ is allowed to ask the adversary for (a) the next neighbor of any vertex (and thereby iterate through its adjacency list) and (b) the in- and out-degree of any vertex.

The adversary will, for arbitrary $k$, construct a graph with the structure shown in Figure 1, that at its core consists of a fully connected DAG $C = \{c_1, \ldots, c_k\}$. Additionally, the graph consists of $k - 1$ vertices $B = \{b_1, \ldots, b_{k-1}\}$; each bridge vertex $b_i$ has exactly one incoming edge from vertex $c_i$. All bridge vertices $b_i$ but one connect with their single outgoing edge to vertex $c_{i+1}$. The remaining bridge vertex $b_i$ connects to a path of $k^2$ vertices $P = \{p_1, \ldots, p_{k^2}\}$ instead, which in turn has an edge to $c_{i+1}$.

Initially it is not fixed, after which bridge vertex this path appears in the graph. Thus in order to identify this connection between path and core, an adversary can force the enumeration algorithm to explore all neighborhoods of the core or to walk the whole path.

Without correct knowledge of this connection, it is not possible to know between which two core vertices the path needs to be placed in the topological ordering. If the algorithm starts with solution parts that place some vertex of the core or a bridge, it needs at least $\frac{k(k-1)}{2} + 2(k - 1)$ steps before it can give the $2k$th solution part that has to place a vertex from the path. Identifying some connection on the path to start building an order from there, on the other hand, requires accessing more than $k$ vertices before the first output.

Note, that the graph has $|V| = k + (k - 1) + k^2$ vertices and $|E| = \frac{k(k-1)}{2} + 2(k - 1) + k^2$ edges and thus constant average out-degree. Vertex $c_1$ has the maximum out-degree in the graph, thus $\Delta^+ = \delta^+(c_1) = k$. □

If we consider a setting for ascending topological ordering enumeration without preprocessing where the source vertices are given, the adversarial input in the proof of Theorem 4 directly shows that the delay of $O(\Delta^+)$ from Theorem 2 is optimal. During a preprocessing phase of $O(n)$ steps, an algorithm could however solve the adversarial instance completely.

With a similar idea, we can however also show that delay in the order of the average degree is not achievable for ascending topological ordering enumeration even if preprocessing time in $O(n)$ is permitted.

**Theorem 5.** *Ascending topological ordering enumeration on a graph with average out-degree $\overline{\Delta^+}$ and maximum out-degree $\Delta^+ \in \omega(\overline{\Delta^+})$ cannot be solved with delay*

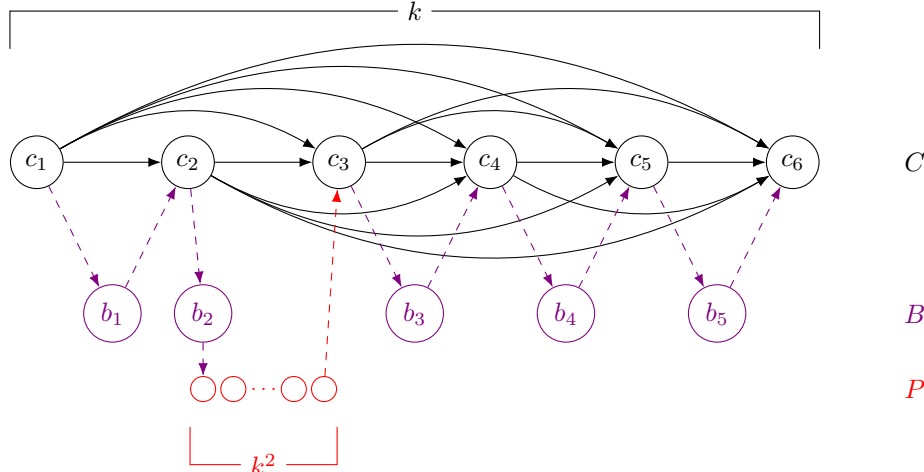

Figure 1: Structure of the adversary input graph.

in $O(\overline{\Delta^+})$, even with $O(n)$ preprocessing and known in-degrees.

*Proof.* Consider, as shown in Figure 2, a graph $G$ of $n = 3k+2+p$ nodes that form a path that completely determines the topological order. Denote the first $k$ vertices as first core group $C_1$, then comes a bridge vertex $b_1$, followed by the second core group of $k$ vertices $C_2$, another bridge vertex $b_2$, the third core group of $k$ vertices $C_3$, followed by a path of $p$ vertices. We will now insert additional edges that, as before, an adversary can use to force an algorithm to inspect too many vertices with high degree before being able to find the path edge: All vertices in $C_1$ connect to all in $C_2$, all in $C_2$ connect to all in $C_3$. Note that except for the source vertex, within each core group all vertices have identical out- and in-degree, so an algorithm cannot distinguish them by degree alone.

We now choose $k = n^{0.6}$. The graph consists of $m = (2k(k + 1) + k) + 2 + (p - 1) = 2n^{1.2} + n - 1$ edges and has an average out-degree of $\overline{\Delta^+} \in \Theta(n^{0.2})$. As the adversary always can force an algorithm to inspect the complete neighborhood of any vertex in $C_1$ and $C_2$, before the algorithm can know the next vertex in the topological order, any algorithm has to perform at least $x \in \Omega(k^2) = \Omega(n^{1.2})$ edge queries to fix the order of two vertices in $C_2$. However, it can up to this point only emit $y \in O(k) = O(n^{0.6})$ solution parts from the first two core groups and $b_1$. With preprocessing in $O(n)$ and a delay in the order of the average out-degree, we get a maximum of $O(n)$ queries the algorithm can execute before running out of solution parts to emit. $\square$

## 4 Application to Scheduling

We now apply the enumeration concept to scheduling problems, starting with variants that can be solved by means of incremental sorting and afterwards applying our algorithm for ascending topological ordering enumeration.

Besides the theoretical analysis we also present experimental results. We implemented all algorithms in Rust and executed the experiments on a compute server with 256 GB RAM and an Intel Xeon Silver 4314 CPU with 2.40 GHz. For each size and parameter we show the average measurements of 10 random instances and 5 runs each. As source for randomness we used the linear congruential generator by Bratley, Fox, and Schrage (1983) as presented in (Taillard 1993). Further details to the instance generation are presented along with the individual problem statements.

### 4.1 Incremental Sorting

It is well known that finding a minimum element in a set of size $n$ requires $\Theta(n)$ steps and that comparison-based sorting is in $\Theta(n \log(n))$. This implies that enumerating the elements of such a set in ascending order requires $\Omega(n)$ preprocessing and $\Omega(\log(n))$ delay, as the first output is the minimum element and there are $n$ solution parts in total. Paredes and Navarro (2006) introduced the INCREMENTALSELECT algorithm that achieves these bounds and emits the $k$th solution part after $O(n + k \log(k))$ steps. The same bounds are met in the average case by their INCREMENTALQUICKSELECT (IQS) algorithm, that performs better in practice.

We apply IQS to the problem of scheduling jobs with release times on a single machine, optimizing the maximum completion time ($1|r_j|C_{\max}$ in standard notation). Scheduling the jobs in order of non-decreasing release time without idle time is optimal, as can be shown with a simple exchange argument. An enumeration algorithm for this problem is expected to produce for each job $j$ a tuple $(j, s_j)$, where $s_j$ is the start time for the job on the single machine. An optimal schedule can be enumerated in order of increasing $s_j$ by keeping track of the maximum completion time $c$ on the machine so far, sorting the jobs according to their release times with IQS, and for each job $j$ emitted by IQS producing the solution part $(j, \max(r_j, c))$.

**Corollary 6.** *An optimal schedule for an instance of $1|r_j|C_{\max}$ with $n$ jobs can be enumerated in order of*

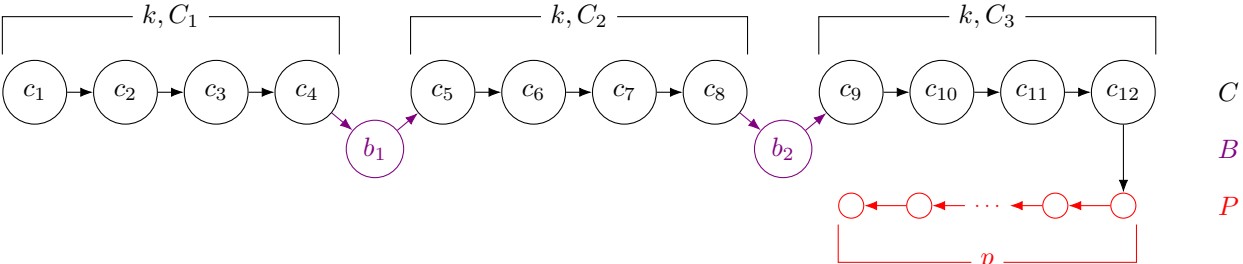

Figure 2: Underlying structure of the adversary input graph for ascending topological ordering enumeration.

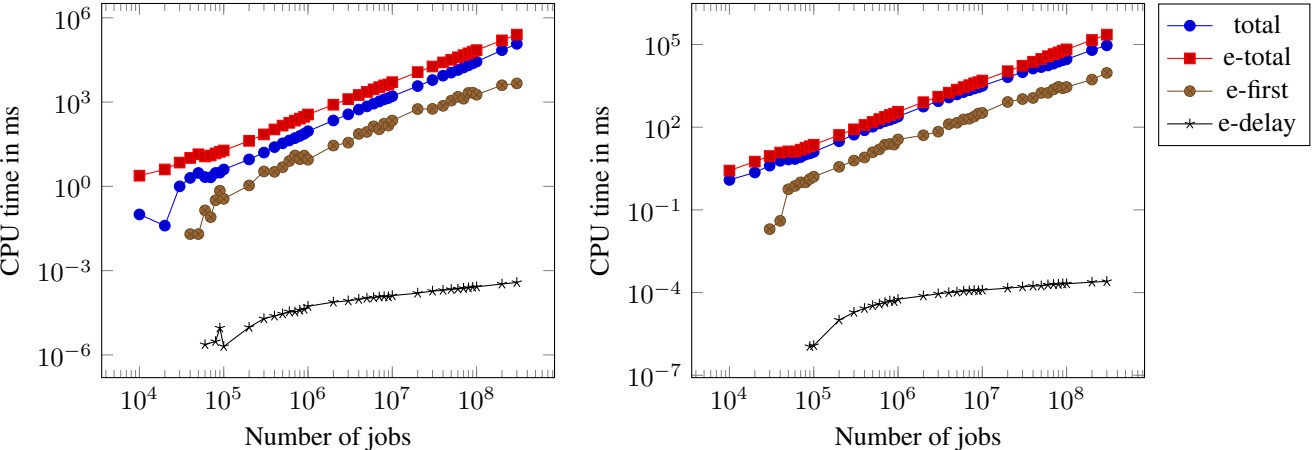

Figure 3: Total time for the classical algorithm compared to the total time, time-to-first-output and average delay of the enumeration algorithm for $1|r_j|C_{\max}$ (left) and $F2||C_{\max}$ (right). Missing data points for e-first / e-delay to the left are 0-values that are dropped in the plot due to the logarithmic scale.

*increasing start times with $O(n)$ preprocessing and with $O(\log(n))$ delay.*

Note that – similarly to sorting – it takes $\Omega(n)$ time to find the potentially single job that can be executed first, which proofs a matching lower bound to the preprocessing time. It is open, however, whether we can also show a matching lower bound on the delay, as sorting by release times is sufficient but not necessary to solve $1|r_j|C_{\max}$ (Lindner et al. 2017).

Corollary 6 easily translates to other problem variants with similar scheduling rules, such as the *earliest due date first* rule for minimizing maximum lateness on a single machine in the presence of deadlines ($1|d_j|L_{\max}$).

Our second example concerns scheduling $n$ jobs in a flow shop with 2 machines: Each job $j$ consists of two operations with processing times $p_{j,1}, p_{j,2}$ and has to be processed first on machine 1 for $p_{j,1}$ time units and later for $p_{j,2}$ time units on machine 2. We again strive to minimize the maximum completion time ($F2||C_{\max}$ in standard notation). This goal is achieved by scheduling all jobs in the same order on both machines, starting with the jobs $j$ with $p_{j,1} \leq p_{j,2}$ ordered by increasing $p_{j,1}$, followed by the remaining jobs in order of decreasing $p_{j,2}$ (Johnson 1954).

The same approach yields an enumeration algorithm that emits for each job and each machine a schedule entry with the respective start time: In the preprocessing phase, the algorithm splits the jobs according to the comparison of processing times on the two machines. It also initializes two IQS instances, one that sorts the first jobs by increasing $p_{j,1}$ and one for sorting the second jobs by decreasing $p_{j,2}$. The enumeration phase consists again of keeping track of the maximum completion times on the two machines so far and scheduling the jobs as emitted by the sorting algorithms. This order already guarantees that schedule entries *per machine* are sorted by start time. In order to sort the entries by start time overall, the enumeration algorithm can buffer the computed solution parts in a queue per machine and, after each delay, emit the solution part from the head of the queue with smaller start time.

**Corollary 7.** *An optimal schedule for an instance of $F2||C_{\max}$ with $n$ jobs can be enumerated in order of increasing start times with $O(n)$ preprocessing and with $O(\log(n))$ delay.*

The similar job shop setting with two machines and at most two operations per job ($J2|n_j \leq 2|C_{\max}$) can be reduced to computing several optimal schedules for $F2||C_{\max}$ (Jackson 1956). Adapted to the enumeration setting this reduction also transfers the time bounds of Corollary 7 to the

more complex job shop.

For our experiments we generated processing times uniformly at random from $\{1, \ldots, 99\}$ (cp. (Taillard 1993)). Release times are chosen uniformly at random from $\{0, \ldots, \frac{T}{2}\}$, where $T$ is the total processing time of all jobs.

Figure 3 shows the runtime measurements. In both experiments we compared the performance of our IQS based enumeration algorithm to a total-time scheduling algorithm based on the pattern-defeating quicksort algorithm from the Rust standard library (Rust Foundation 2023). As expected, the highly optimized standard implementation is roughly two to three times as fast as our unoptimized IQS implementation in the total time comparison. The enumeration algorithm however clearly comes ahead when comparing the time-to-first-output, that is produced after about $\frac{1}{6}$th of the total time of the standard algorithm.

### 4.2 Incremental Topological Ordering

As an application of the ascending topological ordering enumeration, we consider scheduling jobs on a single machine with precedence constraints in the form of a DAG ($1|prec|C_{\max}$ in standard notation). Any schedule without idle time that respects the precedence constraints is optimal in this setting. Thus scheduling in any topological order is sufficient. On top of the algorithm for ascending topological ordering enumeration from section 3 we only have to keep track of the total processing time of the already scheduled jobs to enumerate solution tuples $(j, s_j)$ for each job $j$ and its start time $s_j$.

**Corollary 8.** *An optimal schedule for an instance of $1|prec|C_{\max}$ with $n$ jobs and maximum out-degree $\Delta^+$ can be enumerated in order of increasing start times with $O(n)$ preprocessing and with $O(\Delta^+)$ delay.*

It is possible to extend this algorithm to the slightly more complex scenario with additional release times for the $n$ jobs ($1|prec, r_j|C_{\max}$). A solution algorithm for this setting combines the non-decreasing release time approach of $1|r_j|C_{\max}$ with a topological ordering: The respective next job to be scheduled is one with minimal release time among all available jobs without unfulfilled precedences. We again use the iterative source removal algorithm for the topological order, but do not remove any source, but one with minimal release time. By using a Strict Fibonacci Heap (Brodal, Lagogiannis, and Tarjan 2012) to manage all available source vertices, the algorithm can find and remove such a source in $O(\log(n))$ time and insert the up to $\Delta^+$ new sources after the removal in constant time per insert operation.

**Corollary 9.** *An optimal schedule for an instance of $1|prec, r_j|C_{\max}$ with $n$ jobs and maximum out-degree $\Delta^+$ can be enumerated in order of increasing start times with $O(n)$ preprocessing and with $O(\Delta^+ + \log(n))$ delay.*

We generate the random DAGs for our experiments in the $G(n, p)$ model (Gilbert 1959) by choosing for a fixed vertex order every forward edge with probability $p$ uniformly at random and shuffling the vertices afterwards. Our experiments use, for fast access to both outgoing and incoming edges, two adjacency arrays with vertex offsets as data structure (cp. Kammer and Sajenko 2019).

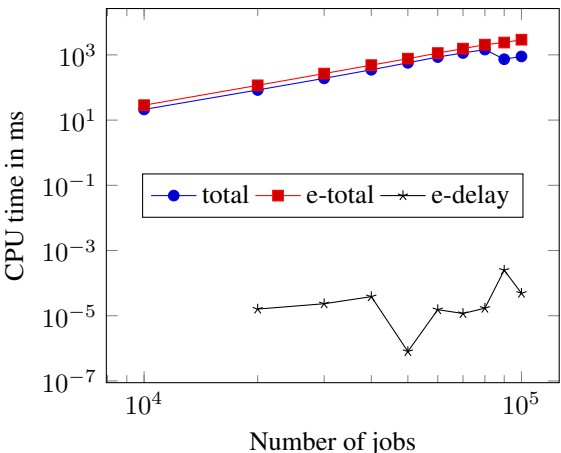

Figure 4: Total time for the total-time algorithm (DFS finish times) compared to the total time and time-to-first-output of our enumeration algorithm for $1|prec|C_{\max}$ with edge probability $p = \frac{1}{4}$.

Figure 4 shows runtime measurements for scheduling jobs based on a dense precedence graph ($p = \frac{1}{4}$) by means of topological ordering. In the total-time comparison, the algorithm based on DFS finish times is again faster than our enumeration algorithm. For the tested instance sizes of up to $10^5$ jobs with roughly $2.5 \cdot 10^9$ edges, the average time-to-first-output measured in milliseconds and averaged with double floating point precision is 0; similarly the measured delay is close to 0. These observations become even more pronounced for sparser graphs.

## 5 Conclusions and Research Directions

In this paper we demonstrated the theoretical and practical potential of an enumeration perspective on scheduling problems. Given the plethora of problem variants in the scheduling literature it is an interesting and wide open question, which variants admit for efficient enumeration algorithms.

Interesting candidates to look at are scheduling problems with a non-trivial polynomial time algorithm. One example is minimizing the number of late jobs on a single machine. The Moore-Hodgson Algorithm (Moore 1968) starts with an *earliest due date first* ordering and from that rejects jobs until the remaining jobs all meet their deadline. It is unclear whether a schedule for this setting could be enumerated efficiently in ascending order, as rejecting jobs is in a way the opposite of fixing and emitting early solution parts.

Another direction might be the comparison of different restrictions to precedence graphs. For example, some scheduling problems seem to profit from the restriction to series-parallel precedence graphs by making use of a series-parallel-decomposition (Lawler 1978). However, a first inspection of such graphs seems to indicate that such a decomposition cannot be enumerated efficiently.

The enumeration concept can also be applied to approximative scheduling. Simple, sorting-based list scheduling rules again transfer nicely to enumeration algorithms: The

LPT rule (*largest processing time first*) for minimizing maximum completion time for parallel machines ($P||C_{\max}$) yields the same approximation ratios as in the offline algorithm (Graham 1969). Better approximation ratios are possible in the offline setting through a reduction to bin packing (Coffman, Garey, and Johnson 1978). Given the offline nature of the reduction it seems unlikely that the same duality holds in the enumeration setting.

Finally, we would like to investigate the gained advantage by enumerating schedules in a processing pipeline. Of particular interest here is the concept of conditional scheduling, where the result of executed jobs determines which subsequent nodes in the precedence graph are to be run and which are discarded (Melani et al. 2015). Whilst basic list scheduling provides acceptable approximation factors in this setting, enumerating a schedule could benefit from a feedback loop: While the schedule is still being enumerated, the already scheduled jobs can run in parallel and their results can then serve as additional input to the scheduling algorithm.

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
