# OpenReview forum: "Incremental Ordering for Scheduling Problems"
_icaps-conference.org/ICAPS/2024/Conference — ICAPS 2024_

### Official Review · Reviewer_aiQp · 2024-01-03

**Significance And Importance:** 2
**Soundness:** 4
**Novelty:** 3
**Clarity:** 4
**Overall Evaluation:** 2
**Confidence:** 4

**Weaknesses:**

2: No major or minor weaknesses.

**Contributions Of The Paper:**

This paper provides algorithms and complexity proofs for incremental enumeration algorithms, which output the first element of the solution after a certain preprocessing time and each successive element after a certain delay (in contrast to traditional algorithms outputting the whole solution at once at the end).
The authors analyze the enumeration complexity (i.e. preprocessing time and delay) of several algorithms: First, they propose enumeration algorithms for topological orderings and show that their enumeration complexity meets the theoretical lower bounds. They then use these algorithms together with incremental sorting from the literature to solve (simple) scheduling problems that can be optimally solved via list-based representations.
Empirical experiments show that while their proposed algorithms require more time than traditional algorithms in total, their time-to-first-solution is significantly shorter.

**Ethical Considerations:**

(1) Not Applicable: The paper does not have any ethical considerations to address

**Nomination For Best Paper:**

No

**Questions For Authors:**

1) Figure 4: Why is the delay fluctuating that much compared to the (near-)monotonic increases in Figure 3? Is it just due to floating point precision issues?

**Reproducibility:**

3: Authors describe the implementation and domains in sufficient detail.

**Strengths Of The Paper:**

The paper is well-written and clearly structured, the provided proofs are technically sound and easy to follow.
This type of enumeration complexity is rarely studied, despite the large amount of work on different scheduling problems, so advances in this area could pave the way for further results, despite the limitation to rather simple scheduling problems.

**Weaknesses Of The Paper:**

Minor issues:
- The term "enumeration complexity" is already used elsewhere (e.g. T’kindt, Vincent, Karima Bouibede-Hocine, and Carl Esswein. "Counting and enumeration complexity with application to multicriteria scheduling." 4OR 3 (2005): 1-21.) for the complexity of counting or enumerating (complete) solutions to a problem
- line 210f: Reference or description of O(n+m) topological sorting missing

Typos:
- line 354: superfluous comma
- line 375: comma should be after "initially" instead of after "fixed"
- line 464: superfluous comma

---

> ### Author Rebuttal · Authors · 2024-01-26
>
> We thank the reviewer for their comments, especially for the suggested reference to a different use of the term "enumeration" related to scheduling problems. We would happily include this differentiation in the introduction.
>
> As for the fluctuating delay in Figure 3: We believe this is due to imprecision in the measurements in milliseconds and the following averaging over several instances and experiment runs. Since the submission we repeated the experiments with nanosecond precision where the odd fluctuations no longer appear, while the overall picture remains unchanged.

---

### Official Review · Reviewer_Cnnx · 2024-01-16

**Significance And Importance:** 1
**Soundness:** 1
**Novelty:** 2
**Clarity:** 2
**Confidence:** 4

**Weaknesses:**

-1: Major weaknesses requiring significant work to be addressed for the paper to be accepted.

**Contributions Of The Paper:**

This paper analyzes an unusual scheduling setup were the goal is to output a schedule for a Directed Acyclic Graph incrementally, the underlying idea being that activities are started while the schedule is being produced. For this reason, the focus on the analysis is characterizing the delays between two consecutive parts of the solution, which intuitively measures how responsive a system of this kind would be in practice.

Given the nature of the considered problem, computing a schedule is trivial once a topological order is know. For this reason, the focus of the authors is on devising and analyzing an algorithm for topological ordering. In particular, the authors analyze the worst-case behavior of the algorithm, and apply it (plus an already existing techniques) to synthetic instances of simple scheduling problems in am empirical comparison.

**Ethical Considerations:**

(1) Not Applicable: The paper does not have any ethical considerations to address

**Nomination For Best Paper:**

No

**Overall Evaluation:**

-2: (reject)

**Questions For Authors:**

See the section about weaknesses.

**Reproducibility:**

2: Some details are missing, but the paper still appears to be replicable with some effort.

**Strengths Of The Paper:**

To the best of my knowledge, the setup considered by the authors is indeed new. While the idea of executing activities while computing a schedule is shared by all forms of online scheduling (e.g. queue based systems, anticipatory algorithms, RL methods), such approaches typically assume that the activities are not a priori know, but subject to uncertainty. In this sense, the paper provides an unusual middle ground, where the activities and their precedences are know, but the schedule still needs to be produced incrementally.

The provided analysis strives to be very rigorous, which might seem unnecessary given the simple nature of the considered scheduling problem, but is in fact a necessary to obtain sound theoretical results.

I appreciated the presence of an empirical evaluation, which is sometimes omitted in works with a theoretical focus, since it allows to get a better grasp of how the proposed methods can work in practice.

**Weaknesses Of The Paper:**

Unfortunately, the work has in my opinion several weakness, mainly in the form of several unclear points. I'll discuss them here in no particular order.

First, the considered setup is not really well defined. In section 2, the authors discuss different forms of enumeration problems, but fail to provide a reference formulation for the problem that they tackle. A rigorous theoretical analysis requires a formal characterization of both the input (I guess it is a DAG) and the output (I guess it is an assignment of start times) of the considered problem (i.e. function). It is of course a good thing if several applications can be tackled by a single approach, but this can be conveyed by showing how these multiple setup can be mapped to the reference formulation.

Second, the connection with online scheduling should in my opinion be better investigated. At the very least, it seems like online scheduling approaches (e.g. queue based system) could be used to address this setup, intuitively with inferior results.

Third, the authors assume that some preprocessing is possible before starting enumeration, i.e. incremental schedule production. It is however very unclear what can be done at preprocessing time and what should instead be considered part of the enumeration. I am also not certain (this might be my fault) whether preprocessing time is taken into account when characterizing the worst case delay. This point is of course critical, since in an extreme case the entire schedule computation could be done at preprocessing time.

Fourth, some information is omitted or not clearly stated. For example, many of the complexity reported for classical problems seems to assume graphs with bounded degree, since only dependency on the number of nodes (rather than edges) is frequently reported.

Similarly, some operation in the core algorithm (Algorithm 1) are not well defined. In particular, if "enqueue" simply represents insertion at the end of a queue data structure it seems to me that the algorithm could lead to repeated nodes and nodes being scheduled before all their predecessors have executed.

Sixth, while the introduction seems to convey the idea that the authors are analyzing the complexity of a problem, but most of the paper is rather focused on Algorithm 1. If the intention is to analyze an algorithm, then this should be clearly stated in the introduction. If the intention is to study a problem, then it should be proved why Algorithm 1 has the best complexity of all algorithms that could be used to solved the problem.

From both points of view, the roles of the IncrementalSelect and IncrementalQuickSelect approaches should be clarified: are these algorithms treated as the SotA for non incremental algorithms? Do they solve the same problem as Algorithm 1, or a different one? Indeed, from a comment on page 7, it seems that the existing algorithms and the proposed one operate under different assumptions, which would make the comparison close to a classical apple-vs-oranges case and reduce its significance.

Finally, given the low complexity of the considered problem and the considered instance sizes, reporting results in terms of run time may not be a robust approach, as caching effects and the action of the System-on-Chip scheduler or frequency governor may have a noticeable impact. Measuring complexity in terms of number of operations, or considering larger instances, would be advised.

---

> ### Author Rebuttal · Authors · 2024-01-26
>
> We thank the reviewer for their comments. First, we want to address the remarks on correctness. In general, we feel to have already covered the mentioned aspects. We are open to suggestions on how to highlight them better:
>
> - 1. Sec. 2 consists of a general introduction, followed by the formal definitions of three different specific problems. This includes the input for the topological ordering enumeration in l. 145ff (a DAG). and the output for schedule enumeration in l. 192ff. (tuples of job, machine, start time).
> - 3. Delay is only measured in the enumeration phase (l. 141f.). The preprocessing phase could encompass the complete computation, defeating the purpose of enumerating parts. Thus, in section 3 we analyze the tradeoff between preprocessing time and delay.
> - 4. All Theorems/Corollaries concerning graphs refer to the number of edges or the out-degree of vertices. We do not assume bounded degrees or any other restriction. The other statements concern sorting numbers.
> - 5. We apologize for the misleading title of Algorithm 1: it only shows how the queue of solution parts is filled (see l. 281ff). DFS visit is only called once per vertex, preventing duplicates. We'd happily add this remark to the proof. Algorithm 1 explicitly works out-of-order; this is the difference between Theorem 3 (topological ordering enumeration) and Corollary 2 (*ascending* t. o. e.).
> - 6. Theorems 3 and 4 prove optimality of our algorithm (l. 345f.).
> - 7. We compare IncrementalQuickSelect to the SotA pattern-defeating quicksort (l. 533ff.). Our algorithm deals with the different problem of topological ordering enumeration; we never compare the two.
>
> The reviewer raises additional important points for discussion:
>
> - 2. Online algorithms can be used for enumerating a schedule if the requirement on the output order is dropped. As suggested, one would expect inferior results. We would be happy to include this remark. A more in-depth comparison would, however, be part of future work.
> - 8. In our theoretic results we prove, that the number of operations for time-to-first solution is better for the enumeration variants. We did implement several mitigations for external influences on the experiments (random order of execution, repeated experiments, multiple instances per size). Still, we agree that even more experiments are an important next step. In our setup, larger instance sizes would lead to more robustness problems due to paging, as larger graphs would no longer fit into RAM.

---

### Official Review · Reviewer_J4xs · 2024-01-23

**Significance And Importance:** 3
**Soundness:** 3
**Novelty:** 3
**Clarity:** 3
**Overall Evaluation:** 2
**Confidence:** 3

**Weaknesses:**

1: Minor weaknesses that are easily fixable.

**Contributions Of The Paper:**

This position paper proposes a new type of scheduling problems where the entire problem is known beforehand and steps in the optimal solution are released as soon as possible, even before finishing the entire search to get a complete solution. The difference of the problem type from online/semi-online scheduling is noted: in online scheduling, the problem is not given beforehand but the future jobs come to the system dynamically and are unknown when the decisions are made for the already arrived jobs. In the proposed incremental scheduling, information about all jobs can be used through some preprocessing steps. The paper discusses a total ordering-type scheduling problems as candidate and proposes to use ascending topological sorting with incremental sorting  as solution approaches that provide upper and lower bounds on complexities. Some theoretical properties are proved and a tiny experiment is performed.

**Ethical Considerations:**

(1) Not Applicable: The paper does not have any ethical considerations to address

**Nomination For Best Paper:**

No

**Questions For Authors:**

Please try to address the last two comments in the Weakness section.

**Reproducibility:**

3: Authors describe the implementation and domains in sufficient detail.

**Strengths Of The Paper:**

1. The paper reads well.
2. I think the work is interesting.
3. It opens up a branch of scheduling although optimality assumption needs to be sacrificed in many cases.
4. The theoretical analysis and the mini experimental results show the essential things.

**Weaknesses Of The Paper:**

1. Adding a concrete small application example would help understand the approach more.
2. A clear demarcation about the sections/materials that are already present and that are proposed would help see the contributory contents more clearly.
3. Not sure whether the related work needs to include more. Greedy algorithms that are proven to give optimal solutions should have the incremental nature.
4. Given that most practical problems are NP-Hard, how restricted is the optimality assumption?
5. The experimental results shows problems needing DFS for up to 10^5 nodes. Considering "optimality" this type of number of be large for many other types of problems. How would the scalability affect the other directions to be explored in the future.

---

> ### Author Rebuttal · Authors · 2024-01-26
>
> We thank the reviewer for their comments and - as they asked - want to address the last two comments in the Weakness section:
>
> 4. It is true, that we only performed experiments for scheduling variants that yield an optimal solution in polynomial time. However, we don't assume optimality; in fact in section 5 (Conclusions and Research Directions) we comment on approximative solutions and give an example for the LPT rule for the NP-Hard problem $P||C_{max}$ (l. 617ff). While (unless P=NP) one cannot hope for short preprocessing time and a polynomial delay for optimal solutions to NP-Hard scheduling problems, future research on enumeration of solution parts to an approximative solution is an important endeavor.
> 5. In the submitted paper we showed the difference primarily for larger graphs, as both the total time and the delay for topological ordering is quite small (linear). Since the submission we re-ran the experiment with nanosecond precision for our time measurements. This allowed us to measure non-zero times for smaller graph instances which confirmed the submitted picture. In general, however, if the total time is very small in practice, enumeration might not be of much interest, as there is no relevant waiting time to improve upon. For harder problems the approach still scales nicely, if the computational effort can be distributed somewhat evenly over all delays. Then, enumeration is valuable for smaller inputs as well.
>
> Finally, we want to agree on the suggestion to investigate more greedy scheduling algorithms for their enumerative behavior. Specifically one has to (1) analyze the order of produced solutions, and (2) optimize potential preprocessing overhead or bad worst-case delays. Both aspects are usually not covered in existing formulations, as they are irrelevant in a total-time perspective on the problem. (cp. the suggestion of reviewer 2 to do the same for existing online algorithms)

---

### Meta-Review · Area_Chair_8Uez · 2024-02-06

**Recommendation:** Accept (Oral)
**Confidence:** 5

**Metareview:**

This paper proposes and analyzes a new type of scheduling problem where, even though the entire problem is known in advance, it is desirable to start executing jobs as soon as possible (i.e., as soon as the first part of the solution can be emitted, and before the algorithm that is generating the solution has completed). In this setting the paper analyzes the “enumeration complexity” of core scheduling problems that reduce to ordering problems, coupling ascending topological ordering with an incremental sorting algorithm to establish upper and lower bounds on complexity. Some theoretical properties are proved, and a small experimental study is performed.

The strengths of the paper include the following. First, the paper identifies an interesting class of previously unstudied scheduling problem. Online scheduling approaches typically assume that the jobs to be scheduled are not known in advance and emerge dynamically as execution proceeds. Second, although the current paper makes rather simplistic assumptions about scheduling problem constraints, it introduces an analysis framework that could be applied to more complex problem formulations. The theoretical analysis is rigorous, and the inclusion of experimental results show how the techniques techniques can work in practice. Third, the paper is well written, and clearly structured. The proofs are technically sound and easy to follow. Overall, the concept of enumeration complexity is rarely studied, and seems much more broadly applicable.

There are also some weaknesses to the paper in its current form. First, the connection with prior research in online scheduling should be given more attention, particularly with respect to queuing models and greedy solution techniques. Second, the paper confuses the notions of analyzing the complexity of a problem and analyzing the complexity of an algorithm for solving a given problem when motivating the work.

The reviews have provided several comments identifying points of confusion and suggestions for improving the presentation of the paper. Please incorporate revisions (including the proposed changes identified in your rebuttal) to address these comments in the final version of the paper.

**Ethical Considerations:**

(1) Not Applicable: The paper does not have any ethical considerations to address